# Procurement Auctions with Best and Final Offers

## Abstract

We study sequential procurement auctions where the sellers are provided with a "best and final offer" (BAFO) strategy. This strategy allows each seller $i$ to effectively "freeze" their price while remaining active in the auction, and it signals to the buyer, as well as all other sellers, that seller $i$ would reject any price lower than that. This is in contrast to prior work, e.g., on descending auctions, where the options provided to each seller are to either accept a price reduction or reject it and drop out. As a result, the auctions that we consider induce different extensive form games and our goal is to study the subgame perfect equilibria of these games. We focus on settings involving multiple sellers who have full information regarding each other's cost (i.e., the minimum price that they can accept) and a single buyer (the auctioneer) who has no information regarding these costs. Our main result shows that the auctions enhanced with the BAFO strategy can guarantee efficiency in every subgame perfect equilibrium, even if the buyer's valuation function is an arbitrary monotone function. This is in contrast to prior work which required that the buyer's valuation satisfies restrictive properties, like gross substitutes, to achieve efficiency. We then also analyze the seller's cost in these subgame perfect equilibria and we show that it can vary significantly across different efficient outcomes, depending on the structure of the buyer's valuation function.

### ACM Reference Format:
Anonymous Author(s). 2024. Procurement Auctions with Best and Final Offers. In *Proceedings of Make sure to enter the correct conference title from your rights confirmation emai (The Web Conference '25)*. ACM, New York, NY, USA, 9 pages. https://doi.org/10.1145/nnnnnnn.nnnnnnn

## 1 Introduction

Many situations call for purchasing items that need to fit together. A computer drive must fit in the drive bay, an alternator in the engine bay, and a piece of accounting software may need to be compatible with inputs from a web store. Indeed, a company setting up a web store needs a variety of software products that work together: a front end for the buyer, an accounting system, an inventory system, a process for product delivery, customer service software, and so on. The creator of a web store will want to purchase various applications that are compatible and, as a result, two accounting systems may be good substitutes, but an accounting system will be a complement to an inventory system. That is, an array of software applications will contain both substitutes and complements, in a potentially very complex relationship.

Procurement problems featuring both substitutes and complements are probably more common than just substitutes. Virtually any assembly problem will force complements via specifications and compatibility. While individual components are substitutes for each other, components that fit together are necessarily complementary. This logic applies equally well to software as well as hardware. Similarly, information – for advertising, for financial transactions, for investment, or for training models – from various sources will have both complements and substitutes among the various suppliers. How should a buyer purchase in such a situation?

The default solution for pricing an asset when there is substantial uncertainty about its value is to hold an auction, and this approach has been successful for a variety of such assets, ranging from electromagnetic spectrum licenses to antiques and art. We approach this problem from the buyer's perspective, aiming to design procurement (or reverse) auctions that they can use to determine which subset of these goods or services they should acquire, and at what price. One difficulty in applying standard procurement auction tools to the applications discussed above is that known techniques from the literature focus on the case where the goods being sold are substitutable, e.g., captured by the assumption that the buyer's valuation function is submodular, or satisfies the more restrictive gross substitutes property. For example, the brilliant paper of Kelso and Crawford [13], extended by Gul and Stachetti [11], shows that whenever the gross substitutes property is satisfied the auctions perform well. Moreover, there are examples that illustrate bad equilibria can arise whenever the gross substitutes property is violated, suggesting that this property may even be necessary.

Our main contribution in this paper is to design auctions that can handle the more complicated value structure that arises in many important applications. We revisit the design of procurement auctions and augment the bidder with a new strategy which we call "best and final offer" (BAFO). This strategy, which is often used in practice but has not received enough attention in auction theory, effectively allows the bidders to freeze their price, while remaining active. In doing so, they risk losing the auction but, as we show, if they strategically choose when to use it, the design recovers the efficiency even in settings with highly complicated valuation functions that exhibit both substitutes and complements.

### 1.1 Our Results

We focus on settings involving a single buyer who wants to acquire the goods or services of multiple sellers. For that setting we revisit the design of procurement auctions and augment the bidder with a new strategy which we call "best and final offer" (BAFO). This strategy, which is often used in practice but has not received enough attention in auction theory, effectively allows the bidders to freeze their price, while remaining active. In doing so, they risk losing the auction but, as we show, if they strategically choose when to use it, the design recovers the efficiency even in settings with highly complicated valuation functions that exhibit both substitutes and complements.

                                                                                               

In the first class of auctions, which we refer to as "Name-Your-BAFO" auctions, the sellers are approached in some sequence and each seller $i$ is asked to directly report their BAFO, i.e., the payment that they request for selling (essentially posting a take-it-or-leave-it price), after observing the prices posted by all sellers that were approached before $i$ in the sequence. Once all sellers have posted their prices, the buyer purchases from the set of sellers that maximizes their utility, i.e., the set that maximizes the buyer's value for the set minus the sum of the prices posted by the sellers in that set.

The second class of auctions are descending auctions and may interact multiple times with each seller, providing them with a strategy space that is quite different. These auctions initially assign a high price to each seller and then take place over a sequence of rounds: in each round the auction computes a tentative allocation based on the current prices and approaches one of the sellers that remain active and is not in the tentative allocation and asks them if they would be willing to accept a slightly reduced price. The seller can either accept that reduction and remain active, or to instead permanently "freeze" their price. Crucially, a seller that freezes their price does not drop out of the auction, so freezing their price corresponds to a BAFO strategy. Once every seller is either frozen or in the tentative allocation, the buyer purchases from the set of sellers that maximizes their utility, given the current prices.

*Analysis of auction efficiency and cost.* We analyze the extensive form games that these sequential auctions give rise to when the sellers know the buyer's valuation function and the sellers' costs (i.e., the bare minimum payment that they need to receive), but the buyer has no prior information regarding the sellers' costs. Specifically we evaluate the subgame perfect equilibria (SPE) of these games in terms of their efficiency and their cost.

We first show that every one of these auctions is guaranteed to reach efficient outcomes in every subgame perfect equilibrium. This result holds for a very general class of buyer valuation functions that can exhibit complementarities, which is in stark contrast to classic prior work that seems to suggest that the restrictive gross substitutes property is required to reach efficiency. The key differences that allow us to achieve this positive result are i) the fact that our auctions provide the sellers with the ability to make a BAFO, combined with ii) their sequential implementation, which allows the sellers to signal to each other using the BAFO strategy. Specifically, once some seller $i$ has finlized their price using the BAFO strategy, all other active sellers can observe this fact and have to choose their optimal strategy conditioned on this fact. This is something that $i$ anticipates when choosing the BAFO strategy, and this signaling between the sellers avoids inefficient outcomes.

Finally, apart from the allocations returned by these auctions we also analyze the price vectors that they give rise to. We show that although the allocation remains efficient irrespective of the sequence in which the sellers are approached, this sequence can have a very big impact on the total cost that the buyer needs to pay, i.e., the sum of the prices in the efficient solution.

## 1.2 Related Work

Our work adds a new twist to the large body of literature on ascending and descending price auctions, which have a long history in economic theory (e.g., Kelso and Crawford [13], Demange, Gale

and Sotomayor [9]; Gul and Stacchetti [11]; Parkes and Ungar [17]; Ausubel and Milgrom [4]; Bikhchandani and Ostroy [6], Ausubel [1, 2]; Perry and Reny [18]; de Vries, Schummer, and Vohra [8]). Such designs have been immensely successful both in theory and in practice and variants of these designs have been used in major spectrum auctions worldwide [3, 16] as well as in auctions for electricity, gas, and emission allowances in Europe [7], among many other applications. There are several practical advantages to this auction format such as minimizing the information the buyer learns about the sellers and the simplicity in bidding (see [18] for a comprehensive discussion).

Prior work has focused on settings in which there is a degree of substitution between the items involved: [11, 13] focus on gross substitutes, [9] on unit-demands which is a special case of substitutes, [1] on homogenous goods with decreasing marginals, [8] assumes a submodularity condition. The only examples that we are aware of with ascending and descending price procedures that can handle complements are: Sun and Yang [20] and Baranov et al [5] who do so by studying restriced forms of complementarity with an underlying substitutable structure. This is a structure that Hatfield and Kominers refer to as "hidden substitutes" [12]. In fact, some of these auctions are known to lose efficiency even in seemingly very simple settings beyond substitutes [10].

Just like the descending auctions that we analyze in this paper, the descending clock auctions in the papers cited above also assign a personalized price to each seller, which then weakly decreases over time. However, a crucial difference is that clock auctions do not provide the bidders with the option of freezing their price. If a seller is not willing to accept a price decrease, then they are forced to drop out of the auction and are, therefore, excluded from the final solution, even if their price before dropping out turns out to be competitive in hindsight.

Our results augment the descending auction format with the ability for participants to make a BAFO – instead of dropping out of the auction, sellers can remain active but can no longer revise their price. This new feature allows us to extend the efficiency guarantees to any combinatorial valuation. The notion of a BAFO has been used in the implementation of optimal strategies in certain bargaining games (e.g. Samuelson [19]) but to the best of our knowledge has not been applied to iterative combinatorial auctions.

Another important difference between our work and the previous literature on iterative auctions is the equilibrium concept. Traditional descending price auctions satisfy the stronger notion of strategyproofness. Instead we guarantee efficiency under any subgame perfect equilibrium (SPE) of the extensive form game induced by the auction mechanism. Characterizing SPE of tends to be difficult except for very structured games [14]. The SPE of simple auction formats has been studied under submodularity and matroid conditions [15].

## 2 Preliminaries

We consider settings with a single buyer who wants to procure goods or services from a set $N = \{1, \ldots, n\}$ of $n$ sellers. Each seller $i \in N$ has a cost $c_i \in \mathbb{N}$ for selling[1] and the buyer's value for buying

---

[1]Throughout the paper, we assume that costs and prices are expressed as multiples of some small enough denomination, e.g., \$1 or ¢1.

each subset of goods or services is captured by a combinatorial valuation function $v(\cdot) : 2^N \to \mathbb{R}$. Given a vector of prices $p \in \mathbb{R}^n$, one for each seller, the utility of the buyer for a subset of sellers $Q$ is equal to his value for that subset, minus the total cost, i.e., $v(Q) - \sum_{i \in Q} p_i$. We use $D(v; p)$ to denote the buyer's preferred subset of sellers, i.e., the one that maximizes the buyer's utility:

$$D(v; p) := \text{argmax}_{Q \subseteq N} [v(Q) - \sum_{i \in Q} p_i].$$

*Types of valuation functions.* A valuation function $v(\cdot) : 2^N \to \mathbb{R}$ is *anonymous* if for every $Q \subseteq N$, the value of $v(Q)$ depends only on the size $|Q|$ of this set, irrespective of the sellers in it. A valuation function is *submodular* if for any two sets $Q$ and $R$ such that $Q \subset R \subset N$, and any $i \notin Q \cup R$, we have $v(Q \cup \{i\}) - v(Q) \geq v(R \cup \{i\}) - v(R)$. A buyer's valuation function satisfies the *gross substitutes* property if an increase of the price for certain goods does not reduce the buyer's demand for goods whose price did not increase. Formally, a valuation function satisfies *gross substitutes* if for every pair of price vectors $p \leq p'$, if $Q \in D(v; p)$ at prices $p$, then there exists $R \in D(v; p')$ at prices $p'$ such that $Q \cap \{i : p_i = p'_i\} \subseteq R$.

*Procurement auctions.* Our goal is to design procurement auctions, which are mechanisms that interact with the sellers and decide which subset $W \subseteq N$ of sellers the buyer will purchase (the "winners"), and what price each winner $i \in W$ should pay. The output of the auction is the set $W$ (or, equivalently, a vector $x \in \{0, 1\}^n$, such that $x_i = 1$ if $i \in W$ and $x_i = 0$ if $i \notin W$) and a price $p_i$ for each $i \in W$. The utility of each seller $i \in N$ is

$$u_i(x, p) = (p_i - c_i)x_i.$$

If $p_i = c_i$ we assume that agents prefer winning at a price equal to their cost to losing (both yield the same utility of zero).

Given a set of winners $W$ and a price vector $p$, the (utilitarian) social welfare is equal to $v(W) - \sum_{i \in W} c_i$. We use $W^* \in \arg\max_{W \subseteq N} v(W) - \sum_{i \in W} c_i$ to denote an *efficient* solution, i.e., a set of winners $W$ that maximizes the social welfare, and we use $x^*$ to denote the corresponding allocation vector. The total cost of the buyer given a price vector $p$ and a set of winners $W$ is $\sum_{i \in W} p_i$.

*Winner selection and tie-breaking.* Once the price vector $p$ has been finalized, our proposed auctions choose a set of winners $W$ from the demand set $D(v; p)$. If there are multiple such sets in $D(v; p)$, we use a tie breaking rule which is encoded as a winner selection rule which maps a vector of prices $p$ to a set of winners $W(p) \in D(v; p)$ in the demand set and satisfies the well-known Independence of Irrelevant Alternatives (IIA) property. In our context, this implies that if $W^*$ is the winning set that we choose from $D(v; p)$ and, $D(v, p')$ is another demand set such that $D(v, p') \subseteq D(v, p)$ and $W^* \in D(v, p')$, then we choose $W^*$ from $D(v, p')$ as well. Most natural ways of breaking ties satisfy IIA. E.g., the rule that defines some arbitrary total order over subsets $Q \subseteq N$ and then chooses the first $Q \in D(v; p)$ according to this total order.

*Extensive form games and subgame perfect equilibrium.* Our auctions are sequential and give rise to extensive form games between the sellers. An extensive form game with a set $N$ of players is represented by a rooted tree $\mathcal{T}$ of finite depth with node set $\mathcal{S}$. Given a node $s \in \mathcal{S}$, we use $\mathcal{T}(s) \subseteq \mathcal{S}$ to denote the child-nodes of $s$ in $\mathcal{T}$. If $\mathcal{T}(s) = \emptyset$, we say that $s$ is a terminal node; otherwise, we say that

it is an internal node. We use $\mathcal{S}_{\text{term}}$ to denote the set of terminal nodes and $\mathcal{S}_{\text{int}} = \mathcal{S} \setminus \mathcal{S}_{\text{term}}$ to denote the set of internal nodes.

Each internal node $s$ is associated with a player $i(s)$ (the player whose turn it is to make a "move" at that point in the game) using a mapping $i : \mathcal{S}_{\text{int}} \to N$. If we let $\mathcal{S}_i \subseteq \mathcal{S}_{\text{int}}$ be the set of internal nodes associated with player $i$, then a strategy for player $i$ is a mapping $a_i : \mathcal{S}_i \to \mathcal{S}$ such that $a_i(s) \in \mathcal{T}(s)$. Each terminal node is associated with a payoff $\pi_i : \mathcal{S}_{\text{term}} \to \mathbb{R}$ for each player $i \in N$. A profile of strategies $(a_1, \ldots, a_n)$ for all players extends the payoff functions from the terminal nodes to every internal node $s \in \mathcal{S}$ via backward induction as follows:

$$\hat{\pi}_j(s) = \pi_j(s) \text{ for } s \in \mathcal{S}_{\text{term}} \text{ and } \hat{\pi}_j(s) = \hat{\pi}_j(a_{i(s)}(s)) \text{ for } s \in \mathcal{S}_{\text{int}}.$$

A profile of strategies is a subgame perfect equilibrium (SPE) if for every player $i$, all nodes $s \in \mathcal{S}_i$ and all nodes $s' \in \mathcal{T}(s)$ we have:

$$\hat{\pi}_i(a_i(s)) \geq \hat{\pi}_i(s').$$

## 3 Sequential Name-Your-BAFO auction

As a warm up, we first propose and analyze the *Name-Your-BAFO*, which is a simpler auction that asks each seller once what their best-and-final-offer is. This auction is simpler to implement and analyze than the descending auction analyzed in Section 4, but lacks the "price discovery" aspect of descending auctions that makes them attractive in practice. These differences notwithstanding, our results show that both of these auctions always yield efficient outcomes.

The Name-Your-BAFO auction approaches the sellers in a pre-specified (and possibly history-dependent) order and sequentially asks each seller $i \in N$ to submit a bid $b_i \in \mathbb{N}$ regarding the price that they would like to be paid for their good or service. This is the only price reported by the seller, so they directly report their BAFO. Before reporting their BAFO, each seller can observe the bids reported by all preceding sellers. The buyer then chooses a set of winners $W$ that maximizes their own utility $v(W) - \sum_{i \in W} b_i$ (using a tie-breaking rule that satisfies IIA) and pays each winner $i \in W$ a price equal to their bid, i.e., $p_i = b_i$.

The auction described above gives rise to an extensive form game that can be represented by a tree of depth $n + 1$, where each node at level $\ell + 1$ is indexed by a tuple $(b_1, \ldots, b_\ell) \in \mathbb{N}^\ell$. Note that the tree has infinite branching but finite depth. The payoffs of the sellers can be computed at the terminal nodes once all the bids $(b_1, \ldots, b_n)$ have been specified. In this section we only refer to this tree implicitly. In the more difficult proofs in the following session, we explicitly analyze the corresponding game tree.

Before analyzing this auction, we consider the toy example of buying chopsticks in auction to develop some intuition regarding the important role of the sequential nature of our auctions.

### 3.1 Buying Chopsticks in Auction

To exhibit the issues that arise with complementarities and how sequential pricing can sidestep these issues, we consider the illustrative example of buying chopsticks using an auction. Consider an instance of a chopstick auction with one seller with a fork and two sellers with one chopstick each. The value of the buyer for a fork is the same as the value for two chopsticks, say \$1; a single chopstick is worthless. This situation fails the gross substitutes property because the chopsticks are complements (raising the price

of one chopstick will eventually lower the demand for the other chopstick). Assume that the chopstick sellers have a cost of ¢10 each, and the fork seller has a cost of ¢50, and consider a non-sequential version of the Name-Your-BAFO auction. In this auction, there is an equilibrium where the chopstick sellers each ask for ¢95 and the buyer buys the fork for $1. No seller can improve their position by a unilateral deviation; a chopstick seller cannot create a sale even if they were to lower their price to equal their cost.

Note, however, that the inefficient equilibrium falls apart if the prices are set sequentially and the sellers can observe these prices. For example, suppose that the auction approaches the sellers in a sequence and the fork seller goes last. If the sum of the chopstick prices exceeds the fork seller's cost, the fork will ask for the sum of the chopstick prices, perhaps minus a penny if the indifferent buyer randomizes (or $1 if the sum of the chopstick prices also exceeds the buyer's value). Knowing this, the second chopstick seller will ensure that the sum of the prices does not exceed the fork's cost, if that is possible (i.e., if the price asked by the first seller is not more than ¢40). Knowing this, the first chopstick asks for exactly ¢40, the second seller asks for ¢10, and the fork seller asks for ¢50. The other cases, where the fork seller is not last, are similar but the fork does not necessarily ask for a price equal cost. In all permutations, the buyer purchases chopsticks, which is the efficient outcome, though the payment may exceed ¢50.

REMARK 3.1. *As exhibited by the chopstick auction example, in the subgame perfect equilibria of the Name-Your-BAFO auction the sellers' bids are not necessarily equal to their costs.*

## 3.2 Analysis of Name-Your-BAFO Auctions

Our main result for this section shows that every subgame perfect equilibrium of a Name-Your-BAFO auction is efficient.

THEOREM 3.2. *For any combinatorial valuation $v$, the allocation induced by any subgame perfect equilibrium of a Name-Your-BAFO auction is efficient.*

Before proving this theorem, we show the following lemma. Specifically, since all auctions in this paper choose an allocation that maximizes the buyer's utility (using a tie-breaking rule that satisfies IIA), the outcome always satisfies the following property.

LEMMA 3.3. *If for some prices $p$ our auction chooses the winning set $W(p)$, and $p'$ are any prices such that $p'_i \leq p_i$ for $i \in W(p)$ (the prices of all the winning bidders are weakly lower) and $p'_i \geq p_i$ for $i \notin W(p)$ (the prices of all the losing bidders are weakly higher), then our auction chooses the same winning set at $p'$, i.e., $W(p') = W(p)$.*

PROOF. Let $W^* = W(p)$ be the set of winners at prices $p$ and let $T \neq W^*$ be any other set of sellers. Since $W^* \in D(v; p)$, we have

$$v(W^*) - \sum_{i \in W^*} p_i \geq v(T) - \sum_{i \in T} p_i. \quad (1)$$

Under prices $p'$, the cost of $W^*$ decreases by $\Delta = \sum_{i \in W^*} p_i - p'_i$ and the cost of every other set $T$ decreases[2] by at most $\sum_{i \in T \cap W^*} p_i - p'_i$,

---

[2]In fact, it may even increase.

which is at most $\Delta$, i.e., $\sum_{i \in T} p'_i \geq \sum_{i \in T} p_i - \Delta$. As a result, we get:

$$v(W^*) - \sum_{i \in W^*} p'_i = v(W^*) + \Delta - \sum_{i \in W^*} p_i$$
$$\geq v(T) + \Delta - \sum_{i \in T} p_i$$
$$\geq v(T) - \sum_{i \in T} p'_i,$$

where the first inequality is due to (1) and the second inequality is due to the fact that the prices in $T$ drop by at most $\Delta$. Therefore, $W^*$ is also in $D(v; p')$ (the demand set based on the new prices $p'$) and $D(v; p') \subseteq D(v; p)$ (the demand set based on $p'$ contains only the sets of sellers that were in $D(v; p)$ and whose price dropped by exactly $\Delta$). Using the fact that our auctions use a tie-breaking rule that satisfies the IIA property, this implies that since $W^*$ was selected from $D(v; p)$, it will also be selected from $D(v; p')$. □

We now use this lemma to analyze the subgame perfect equilibria of the game induced by Name-Your-BAFO, which can be computed using backwards induction. First, we focus on the bid of the last seller given the prices posted by the $n - 1$ bidders preceding it, then we focus on the bid of the next-to-last seller, and so on.

For the last seller, choosing their optimal BAFO is relatively straightforward, since all the other bidders' prices are already finalized. The bidder chooses their bid $b_n$ aiming to maximize their utility $(b_n - c_n)x_n$, where $x_n = 1$ if the last bidder is in the winning set and $x_n = 0$ otherwise. Note that, according to the definition of the auction, if the bidder wins, the price that they are paid is $p_n = b_n$, but the winning set $W(p)$ chosen based on the final prices and, hence, also the value of $b_n$ that maximizes their utility, depends on the previously posted prices $(p_1, ..., p_{n-1})$.

To simplify our analysis for sellers that arrive earlier in the ordering, we now define a "conditional price vector" for each round of the Name-Your-BAFO auction.

DEFINITION 3.1. *For each round $k \in \{0, 1, \ldots, n\}$ of the Name-Your-BAFO auction, the* conditional price vector *is defined as:*

$$\hat{p}_i(k) = \begin{cases} b_i, & \text{if } i \leq k \\ c_i, & \text{otherwise.} \end{cases}$$

*The final price vector after the completion of the auction is $p = \hat{p}_n$.*

Now, using this price vector, we define a notion of "conditional efficiency." The auction directly associates a winning set $W(s)$ with each terminal node $s$ of the game tree (the set of winners if that is indeed the outcome of the auction). Given a subgame perfect equilibrium, this allocation can also be extended to every internal node $s$, using the subgame perfect equilibrium outcome of the subgame rooted at $s$ (i.e., $W(s)$ is the set of winners in the subgame perfect equilibrium outcome of the subtree rooted at $s$).

DEFINITION 3.2. *Consider a node $s$ at the $k$-th level of the Name-Your-BAFO game tree, after the first $k - 1$ sellers have named their best and final offers $b_1, \ldots, b_{k-1}$. We say that the set $W(s)$ associated with $s$ is* conditionally efficient *if it maximizes the buyer's utility with respect to the price vector $\hat{p}(k - 1)$, i.e.,*

$$W(s) \in \arg\max_X v(X) - \sum_{i \in X} \hat{p}_i(k - 1).$$

*If there are multiple such allocations, $W(s)$ is the one chosen by the same tie-breaking rule used by the auction to determine the winners.*

We are now ready to prove Theorem 3.2.

PROOF OF THEOREM 3.2. We prove that in any subgame perfect equilibrium, the set $W(s)$ associated with each node $s$ of the Name-Your-BAFO game tree is conditionally efficient. We start at level $k = n$ and proceed by backwards induction. This implies that the auction is also conditionally efficient at $k = 1$, for which $\hat{p}_i = c_i$ for all $i \in N$, so the Name-Your-BAFO auction is efficient in every SPE.

The rest of the proof verifies the conditional efficiency for all $k$.

**Base case ($k = n$):** Given any node $s$ at level $k = n$ of the game tree, we consider two possibilities based on whether or not there is a bid $b_n \geq c_n$ that bidder $n$ can make to become a winner given the previously posted bids $(b_1, \ldots, b_{n-1})$. If no such bid exists, then bidder $n$ is not in $W(s)$ (i.e., they are not one of the winners in the SPE outcome of the subgame rooted at $s$) because for this bidder to become a winner they would have to report a bid $b_n < c_n$ that would return negative utility $b_n - c_n$. For every other bidder and, hence, also for every $i \in W(s)$, we have that their conditional price is the same as their final price, i.e., $\hat{p}_i(n-1) = p_i$, and thus $W(s)$ is conditionally efficient. If, on the other hand, there exists some bid $b_n \geq c_n$ that would make bidder $n$ a winner, then the bidder's optimal strategy is to report such a bid in order to be one of the winners (in fact, the bidder's optimal strategy is to report the largest $b_n$ for which they remain a winner). This means that that bidder $n$ will be a winner in the SPE of the subgame rooted at $s$, so this bidder is in $W(s)$. Furthermore, this implies that $W(s)$ is conditionally efficient in this case as well: since i) $W(s)$ maximizes the buyer's utility with respect to the final prices $p$ and ii) the prices $\hat{p}(n-1)$ satisfy $\hat{p}_i(n-1) \leq p_i$ for all $i \in W(s)$ and $\hat{p}_i(n-1) = p_i$ for all $i \notin W(s)$, we can use Lemma 3.3 to conclude that $W(s)$ would also maximize the buyer's utility with respect to $\hat{p}(n-1)$.

**Induction step:** Now, consider any level $k < n$ and assume that for any subgame perfect equilibrium, any level $\ell \in \{k+1, \ldots, n\}$, and any node $s'$ at level $\ell$, we have that $W(s')$ is conditionally efficient. We consider any node $s$ at level $k$ and we consider two possibilities based on whether or not there is a bid $b_k \geq c_k$ that bidder $k$ can make to become a winner in the corresponding child node $s'$ (the child node of $s$ that corresponds to choosing strategy $b_k$) given the previously posted bids $(b_1, \ldots, b_{k-1})$.

If no such bid exists, then bidder $k$ is not in $W(s)$ because for this bidder to become a winner they would have to report a bid $b_k < c_k$ that would return negative utility $b_k - c_k$. For every other bidder and, hence, also for every $i \in W(s)$, we have that their level-$k$ conditional price in $s$ and their level-$(k+1)$ conditional price in any child node of $s$ is the same (either equal to their cost or equal to their final price), and thus $W(s)$ is conditionally efficient.

If, on the other hand, there exists some bid $b_k \geq c_k$ that would make bidder $k$ a winner, then the bidder's optimal strategy is to report such a bid in order to be one of the winners. This means that that bidder $k$ will be a winner in the SPE of the subgame rooted at $s$, so this bidder is in $W(s)$. Furthermore, this implies that $W(s)$ is conditionally efficient in this case as well. Let $s'$ be the child node of $s$ that bidder $k$ chooses in the SPE and note that i) $W(s)$ maximizes the buyer's utility with respect to the level-$(k+1)$ conditional prices at node $s'$ and ii) level-$(k+1)$ conditional price of bidder $k$ at node

$s'$ is weakly greater than the level-$k$ conditional price of bidder $k$ at node $s$ (because the former is $b_k$ while the latter is equal to $c_k$, and $b_k \geq c_k$). All other prices are the same so, using Lemma 3.3 we conclude that $W(s)$ would also maximize the buyer's utility with respect to $\hat{p}(k)$.

□

REMARK 3.4. *The proof actually provides a partial computation for equilibria. First, for any $k$ such that $x_k^* = 1$, where $x^*$ maximizes $v(x) - cx$, $k$ will choose the highest price for which $\max_x v(x) - (p_1, \ldots, p_{k-1}, p_k, c_{k+1}, \ldots, c_n)x$ results in $x_k = 1$. When $x_k^* = 0$, one choice that always results in an equilibrium is $p_k = c_k$, but there could be higher prices that also result in equilibria.*

REMARK 3.5. *When we specify the game, we need to specify a (possibly adaptive) order in which the buyer will approach sellers and that order becomes common knowledge among the sellers who then use it to compute their SPE. Nevertheless, the proof of Theorem 3.2 shows that the SPE depends only on who are buyers that arrived before and what was their BAFO. A remarkable property of this game, is that sellers are able to compute their strategies without knowing the order in which the buyer will approach the remaining sellers.*

REMARK 3.6. *Note that our proof of the auction efficiency does not make any assumptions regarding the order in which the bidders are approached, so it holds for any possible, even adaptive, ordering. As we show in Section 5, this is not true for the prices that the auction returns, which can vary significantly, depending on the ordering.*

## 4 Descending Auctions with BAFO

While the Name-Your-BAFO auction has efficient equilibria, it may be a demanding auction format for the perspective of the sellers in practice. They interact with the auction only once and, in this one interaction, they need to choose one of infinitely many strategies: a bid $b_i \in \mathbb{N}$. As an alternative format, we now present a class of descending auction where the sellers have repeated interactions with the auction as it gradually reduces the prices offered to each of them, and in each interaction a seller needs to choose between just two strategies: to either accept a price decrease or to permanently freeze their price (make a BAFO). This option for a seller to freeze their price is a feature that is often used in practice, but has not received as much attention from an analytical standpoint.

*Class of descending auctions.* A vector of prices $p$ is initialized at a very high level, say for every $i \in N$ we have $p_i = h$ where $h \in \mathbb{N}$ is some arbitrarily large value, and also initializes the set $F$ of "frozen" sellers to be empty The auction then takes place over a sequence of rounds, and in each round the auction computes a (possibly empty) tentative allocation $W(p)$ based on the current prices, chooses a seller $i \notin W(p) \cup F$ who remains active (i.e., has not frozen their price) and provides them with two options: (i) accept a decrease of their price[3] from $p_i$ to $p_i \leftarrow p_i - 1$, or (ii) freeze their price at $p_i$. If a bidder freezes at $p_i$, this means that this is their best and final offer, so the bidder is added to set $F$ and the auction never attempts to lower their price again in the future. Each bidder's decision to accept a reduced price or freeze is observed by every other bidder,

---

[3]Note that we assumed that all costs are natural numbers, using some small enough monetary denomination, like $ or ¢, so we can safely restrict our attention to price decrements of 1. We could, alternatively, just choose a small enough decrement of $\epsilon$.

so both the current prices and the subset of bidders whose prices are frozen (i.e., the set $F$) is public knowledge. There are no restrictions on the order in which the sellers are approached. In particular, it can be history dependent, i.e., dependent of the sellers' choices along the way. The auction terminates after all sellers not in the tentative allocation $W(p)$ have frozen their prices. We assume that if a price reaches zero, the seller automatically freezes. Once all prices are finalized, the set of winners is determined to be the set that maximizes the buyer's utility at the given prices. If there are multiple such sets, the auction uses a tie-breaking rule that satisfies IIA.

*Extensive form game tree.* Each descending auction from the class above induces an extensive form game which can be represented as a tree, with each node corresponding to the interaction of the auction with some seller $i$, asking them to either accept a reduction of their price by 1 or permanently freeze it. Note that in each round of this game one of the prices is either frozen or decreased by an 1 and no price can drop below zero, so the game is finite. For each node $s$ of this tree, we use $p(s)$ to denote the price vector at the time when the auction reaches this node $s$ and we use $F(s)$ to denote the set of sellers that have already chosen to freeze, i.e., they chose to freeze at some node on the path from the root to node $s$. Note that $p(s)$ and $F(s)$ are both fully determined by the path from the root to $s$, as the edges of this path determine when each price is decreased or frozen. If, at node $s$, the seller $i$ who was asked to reduce their price accepts this reduction, then we proceed to the left child-node of $s$, which we denote as $s_\ell$, and if $i$ does not accept this reduction and instead chooses to freeze, then we proceed to the right child-node of $s$, which we denote by $s_r$. Also, let $G(s)$ be the subgame that corresponds to the substree rooted at $s$.

Note that, since the sequence in which the sellers are approached by the descending auction can depend in non-trivial ways on the observed strategic choices, this can lead to an unpredictable trajectory for the price vector. At first glance, this suggests that the induced game would be very demanding for the sellers to play, but our key lemma (Lemma 4.1) shows that sellers do not need to know anything regarding the future price trajectory or details regarding the history to compute their optimal strategy. They only need access to $p(s)$ and $F(s)$.

## 4.1 Analysis of Descending Auctions with BAFO

Given a descending auction and some problem instance, consider any subgame perfect equilibrium of the game tree induced by this auction. We annotate each node $s$ in the game tree with the final allocation $W(s)$ resulting from playing the given subgame perfect equilibrium of the subtree rooted at $s$.

DEFINITION 4.1. *For each node $s$ in the descending auction game tree, we define a price vector $\hat{p}(p(s), F(s))$, or just $\hat{p}(s)$, such that*

$$\hat{p}_i(s) = \begin{cases} p_i(s), & \text{if } i \in F(s) \\ c_i, & \text{otherwise.} \end{cases}$$

We show the following lemma using backwards induction:

LEMMA 4.1. *For every node $s$ of the descending auction game tree, the allocation $W(s)$ resulting as the subgame-perfect equilibrium of*

$G(s)$ *is an allocation that maximizes the buyer's utility with respect to cost vector $\hat{p}$, i.e., an allocation*

$$W(s) \in \arg\max_X v(X) - \sum_{i \in X} \hat{p}_i(s).$$

*If there are multiple such allocations, $W(s)$ is chosen using the same tie-breaking rule that the auction uses to determine the set of winners.*

PROOF. To verify that the lemma holds for every leaf of the game tree, we observe that at every leaf node $s$ all prices have been permanently frozen, so the vector $p$ of prices is finalized. The buyer's chosen allocation in response to these prices would be the allocation $X$ that maximizes $v(X) - \sum_{i \in X} p_i(s)$. Since every seller is frozen at $s$, i.e., $F(s) = N$, we have $\hat{p}_i(s) = p_i(s)$ for all $i$, so the chosen allocation also maximizes $v(X) - \sum_{i \in X} \hat{p}_i(s)$. If there are multiple such allocations, our auction is designed to consistently tie-break, so the lemma holds for all leaves.

Now, consider an internal node $s$ of the game tree and assume that the lemma holds for all of its descendants. Let $i$ be the seller who, at node $s$, is asked to either accept a price decrease from $p_i(s)$ to $p_i(s) - 1$ (leading to child-node $s_\ell$) or to freeze at $p_i(s)$ (leading to child-node $s_r$). By our inductive assumption, if $i$ accepts the price decrease, the resulting allocation will be

$$W(s_\ell) \in \arg\max_X v(X) - \sum_{i \in X} \hat{p}_i(s_\ell), \tag{2}$$

and, if $i$ chooses to freeze, the resulting allocation will be

$$W(s_r) \in \arg\max_X v(X) - \sum_{i \in X} \hat{p}_i(s_r), \tag{3}$$

and in both cases any ties are broken using the same tie-breaking rule.

Now, to identify seller $i$'s optimal strategy at node $s$ we consider four cases based on whether $i$ wins or loses in the aforementioned allocations $W(s_\ell)$ and $W(s_r)$:

- **Case 1: Agent $i$ loses at $W(s_\ell)$.** Note that if this is the case, then seller $i$ also loses at $W(s_r)$. To verify this fact, note that $\hat{p}_i(s_\ell) = p_i(s) - 1$ and $\hat{p}_i(s_r) = p_i(s)$, so $\hat{p}_i(s_r) > \hat{p}_i(s_\ell)$, while $\hat{p}_j(s_r) = \hat{p}_j(s_\ell)$ for all other sellers $j \neq i$. Therefore, for every winner $j \in W(s_\ell)$ we have $\hat{p}_j(s_r) = c_j(s_\ell)$ and every loser $j \notin W(s_\ell)$ we have $\hat{p}_j(s_r) \geq c_j(s_\ell)$. From the statement of the lemma we can conclude that $W(s_r) = W(s_\ell)$, which means that the allocation $W(s)$ resulting from playing the subgame-perfect equilibrium of $G(s)$ is independent of $i$'s strategic choice at node $s$ and, hence, $W(s) = W(s_\ell) = W(s_r)$.

  Since $W(s) = W(s_\ell)$, to prove that the lemma holds for node $s$ as well, it suffices to show that

  $$W(s_\ell) \in \arg\max_X v(X) - \sum_{i \in X} \hat{p}_i(s), \tag{4}$$

  and that this allocation would be chosen in case of ties. To verify that both of these are true, note that $\hat{p}_j(s) = \hat{p}_j(s_\ell)$ for every seller $j$, since no additional freezing took place between $s$ and $s_\ell$. Therefore, our inductive assumption that the lemma holds for node $s_\ell$ directly implies that the lemma also holds for $s$.

- **Case 2: Agent $i$ wins at both $W(s_\ell)$ and $W(s_r)$.** If this is the case, then seller $i$ would choose to freeze, since they would win in both cases (due to the inductive hypothesis), but the price that $i$ would receive if they freeze is higher (it will be exactly $p_i(s)$ if

they freeze, while it would be at most $p_i(s) - 1$ if they do not). Therefore, given this strategic choice of $i$, the resulting allocation $W(s)$ of $G(s)$ will be the same as $W(s_r)$.

Since $W(s) = W(s_r)$, to prove that the lemma holds for $s$ as well, it suffices to show that

$$W(s_r) \in \arg\max_X v(X) - \sum_{i \in X} \hat{p}_i(s), \tag{5}$$

and that $\tau$ would choose this allocation in case of ties. We can verify that (5) holds by using the fact that $W(s_r)$ satisfies condition (3) and then observing that the only difference between $\hat{p}(s_r)$ and $\hat{p}(s)$ is the fact that $\hat{p}_i(s_r) \geq c_i(s)$ because in $s_r$ seller $i$ froze at a price at least $c_i$. Therefore, for each set that does not contain $i$ their "cost" relative to $\hat{p}$ is the same between $s$ and $s_r$, while the "cost" relative to $\hat{p}$ of all sets that include $i$ dropped by the same amount. The fact that $i \in W(s_r)$ implies that $W(s_r)$ satisfies (5). To also verify that $\tau$ would choose $W(s_r)$ in case of ties with respect to $\hat{p}(s)$, note that $c_j(s_r) \leq c_j(s)$ for all $j \in W(s_r)$ and $c_j(s_r) \geq c_j(s)$ for all $j \notin W(s_r)$, so the fact that $\tau$ chose $W(s_r)$ given $\hat{p}(s_r)$ (by our inductive assumption) implies that $\tau$ would also choose $W(s_r)$ given $\hat{p}(s)$ (by definition of the tie-breaking rule). This implies that the lemma holds also for $s$.

- **Case 3A: Agent $i$ wins at $W(s_\ell)$ for a price less than $c_i$ and loses at $W(s_r)$.** In this case, if seller $i$ accepted the price decrease they would end up winning, but for a price that is strictly lower than their cost, leading to negative utility. They would instead prefer to freeze at price $p_i(s)$ and lose in order to maintain a non-negative utility, so the resulting allocation is $W(s) = W(s_r)$. Since $W(s) = W(s_r)$, to prove that the lemma holds for $s$ as well, it suffices to show that

$$W(s_r) \in \arg\max_X v(X) - \sum_{i \in X} \hat{p}_i(s),$$

and that the tie-breaking rule would choose this allocation in case of ties. We can once again verify that this is true using the fact that $W(s_r)$ satisfies condition (3), combined with the facts that $\hat{p}_i(s_r) \geq \hat{p}_i(s)$ for seller $i$ who is not in $W(s)$, while $\hat{p}_j(s_r) = \hat{p}_j(s)$ for all other sellers $j \neq i$.

**Case 3B: Agent $i$ wins at $W(s_\ell)$ for a price of at least $c_i$ and loses at $W(s_r)$.** In this case, seller $i$ prefers the outcome of winning at $W(s_\ell)$ for a price that would give them non-negative utility rather than losing at $W(s_r)$, which would give them zero utility. As a result, they would accept the price decrease and $W(s) = W(s_\ell)$. Since $W(s) = W(s_\ell)$, to prove that the lemma holds for $s$ as well, it suffices to show that

$$W(s_\ell) \in \arg\max_X v(X) - \sum_{i \in X} \hat{p}_i(s),$$

and that $\tau$ would choose this allocation in case of ties. We can once again verify that this is true using the fact that $\hat{p}_j(s) = \hat{p}_j(s_\ell)$ for every seller $j$, since no additional freezing took place between $s$ and $s_\ell$. Therefore, our inductive assumption that the lemma holds for node $s_\ell$ directly implies that the lemma also holds for $s$. □

Using Lemma 4.1, we can now verify that the descending auction is guaranteed to be efficient in any subgame-perfect equilibrium.

**Theorem 4.2.** *The allocation induced by any descending auction in any subgame perfect equilibrium is always efficient.*

**Proof.** Let $s$ be the root node of the game tree. Since no seller has had a chance to freeze at that point, i.e., $F(s) = \emptyset$, we have $\hat{p}_i = c_i$ for all $i$. Using Lemma 4.1 for $s$ we can conclude that the allocation $W(s)$ resulting from a subgame-perfect equilibrium satisfies

$$W(s_\ell) \in \arg\max_X v(X) - \sum_{i \in X} c_i(s),$$

which implies that it is an efficient allocation. □

**Remark 4.3.** *Similar to Remark 3.5 the proof Theorem 4.2 shows that sellers don't need to know the order in which the buyer will approach sellers to compute their SPE.*

**Remark 4.4.** *If the valuation function satisfies gross substitutes, then the final allocation never includes a seller that froze their price. By the definition of subtitutability, if an item is not demanded at a given price vector, it is not demanded at any vector where every other price is weakly smaller. Hence, for the special case of substitutes, our auction behaves exactly like the procedure of Kelso and Crawford [13].*

## 5 Price Vectors Supporting the Efficient Solution

Having shown that the SPE of these auctions are always efficient, we now analyze their performance in terms of the price vectors that support these efficient allocations. As a warm-up, we first provide a small example that verifies these prices are not unique; not even with respect to their sum, i.e., the buyer's total cost. We prove this for the Name-Your-BAFO auctions, but note that this directly extends to the descending auction with BAFO if we approach the sellers in the same order and lower the price of each seller until they freeze.

**Claim 5.1.** *Executing the Name-Your-BAFO auction with different orderings of the sellers leads to the same efficient allocation in every subgame perfect equilibrium but the resulting prices need not sum up to the same amount.*

**Proof.** Consider an instance with three sellers $\{1, 2, 3\}$ such that the buyer's value for its subsets is $v(\{1\}) = v(\{3\}) = 10$. $v(\{1, 2, 3\}) = 20$, and $v(S) = 0$ for any other subset $S$. Let the costs of all sellers be 0. Then, $x^* = \{1, 2, 3\}$ and the order in which our algorithm raises the sellers' prices actually matters: if the algorithm starts by raising the price of seller 2, it will stop when the price is equal to 10 and the prices of sellers 1 and 3 cannot be raised after that point, leading to price vector $p = (0, 10, 0)$. If, on the other hand, our algorithm starts by raising the price of seller 1, it will stop when that agent's price is equal to 10, but it can then also raise the price of seller 3 by the same amount, leading to a price vector of $p' = (10, 0, 10)$. In particular, when raising the price of sellers who are also part of competing allocations allows the prices to be raised further, since raising those prices also hurts the competition. □

Now, moving to a more complicated example, we show that even for the special class of anonymous valuations, that depend only on the number of sellers rather than who these sellers are, the total cost of the buyer can vary by a factor that grows linearly with the number of sellers.

**Theorem 5.2.** *Even if the valuation function of the buyer is assumed to be anonymous, the sum of the prices of two distinct price*

*vectors that arise from executions of the Name-Your-BAFO auction with different seller orderings can vary by a factor $n/2$, where $n$ is the total number of sellers. This bound is tight. On the other hand, if the anonymous valuation function is also weakly concave, then the price vector that supports $x^*$ as a SPE equilibrium is unique (a threshold price for all winners).*

Proof. Consider an instance with n sellers where the value of the buyer for any set of up to $n - 2$ sellers is equal to the size of the set, i.e., additive with value 1 per seller. However, the value of any set of $n - 1$ sellers has value $n - 2$ as well, while the set $x^*$ that includes all agents has value $n - 1$. The cost of every seller is zero.

We first claim that the price vector $(0, 0, \ldots, 0, 1)$ supports $x^*$ as a SPE equilibrium. The utility of $x^*$ under these prices would be $n - 2$ (a value of $n - 1$ minus a total payment of 1) and if any seller $i$ were to raise their price, the utility of $x^*$ would drop below $n - 2$, while the set $x^* - \{i, n\}$ would have a value of $n - 2$ and a cost of 0, maintaining a utility of $n - 2$ and leading to an inefficient outcome of $x^* - \{i, n\}$ instead of $x^*$. (note that even if $i = n$ the set $\{i, n\}$ equals $\{n\}$ and the above still holds)

We now claim that the price vector $(1/2, 1/2, \ldots, 1/2, 1/2)$ also supports $x^*$ as a SPE equilibrium. The utility of $x^*$ under this price vector would be $n/2 - 1$ (a value of $n - 1$ and a total payment of $n/2$), while the utility of any other subset would be at most $n/2 - 1$ (a set of $n - 1$ sellers has value $n - 2$ and price $(n - 1)/2$, leading to a utility of $n/2 - 3/2$, and a set with $k \leq n - 2$ sellers has value at most $k$ and price exactly $k/2$, leading to a utility of $k - k/2 = k/2 \leq n/2 - 1$). If any seller $i$ were to raise their price, the utility of $x^*$ would drop below $n/2 - 1$, while the set $x^* - \{i, j\}$, where $j$ is any seller other than $i$, would have a value of $n - 2$ and a cost of $(n - 2)/2$, leading to utility at least $(n - 2)/2 = n/2 - 1$.

**Tightness:** To prove that the bound of $n/2$ is tight, given any instance let $p$ and $p'$ be any two price vectors that support $x^*$ as a SPE equilibrium. We will prove that the sum of prices in $p'$ cannot be more than a factor $n/2$ greater than the sum of the prices in $p$. In fact, if we let $k$ denote the size of $x^*$, we will prove that the factor can be no more than $k/2$. Our argument uses the fact that given any price vector $p$, if we order the sellers in weakly increasing order of their prices, the solution that maximizes the buyer's utility is a prefix of that ordering.

Assume that $x^*$ contains k sellers and let $p_1, p_2, \ldots, p_k$ be the $k$ smallest prices in $p$, in weakly increasing order. The fact that $p$ supports $x^*$ as a SPE equilibrium means that if we were to raise $p_1$ by an arbitrarily small amount, then the seller whose price we raised would become a loser. Let $k'$ be the number of winning sellers in the new solution after this change in $p_1$.

Case 1: If $k' >= k$ even though the seller whose price was raised is now a losing bidder, this must mean that the price of this seller is now greater than the $k'$ smallest ones in $p$ and, hence, $p_1$ must have been equal to $p_k + 1$, i.e., equal to the smallest price among the original losing sellers. However, since $p$ and $p'$ support $x^*$ as a SPE equilibrium, the prices of each losing seller are the same in $p$ and $p'$ and this means that all the prices that $p'$ assigns to the $k$ winning sellers are all at most $p_k + 1$ and, therefore, also at most $p_1$ (since we just argued that $p_1 = p_k + 1$ in this case), which is the smallest price in $p$. As a result, this would imply that the minimum price in $p$ is at least as large as the maximum winning price in $p'$,

so the total price in $p$ would be at least as high as the total price in $p'$.

Case 2: If $k' < k$, i.e., the size of the solution after raising $p_1$ is strictly smaller, using the fact that the new solution contains the sellers with prices $p_2 + p_3 + \ldots + p_{k'+1}$, we can infer that

$$p_1 + p_{k'+2} + p_{k'+3} + \cdots + p_{k-1} + p_k = v(k) - v(k'),$$

i.e., the marginal value of the last $k - k'$ sellers in $x^*$ was equal to the sum of their prices (since the seller with price $p_1 + \epsilon$ was rejected, it must mean that their price was already among the highest $k - k'$ up to tie-breaking). But, for any other price vector $p'$ to support $x^*$, it must also be the case that the k-k' highest prices must be upper bounded by v(k)-v(k'), otherwise dropping the corresponding sellers would lead to higher buyer utility. Therefore:

$$p'_{k'+1} + p'_{k'+2} + p'_{k'+3} + \cdots + p'_{k-1} + p'_k \leq v(k) - v(k'),$$

Combined, with the previous equation, this yields:

$$p'_{k'+1} + p'_{k'+2} + p'_{k'+3} + \cdots + p'_{k-1} + p'_k \leq p_1 + p_{k'+2} + p_{k'+3} + \cdots + p_{k-1} + p_k.$$

For $k' = k - 1$, this implies that $p'_k \leq p_1$, which would once again lead to the conclusion that the total price in $p$ is at least as high as the total price in $p'$. For $k' = k - 2$, the inequality becomes

$$p'_{k-1} + p'_k \leq p_1 + p_k \Rightarrow$$

$$\frac{2}{k} \sum_{i=1}^{k} p'_i \leq p'_{k-1} + p'_k \leq p_1 + p_k \leq \sum_{i=1}^{k} p_i \Rightarrow$$

$$\sum_{i=1}^{k} p'_i \leq \frac{k}{2} \sum_{i=1}^{k} p_i,$$

where the first derivation used the fact that $p'_{k-1}$ and $p'_k$ are the top two prices among the $k$ winners in $p'$ and thus also at least as high as two times the average price among the winners. In general, if we use $d$ to denote the drop in the size of the winning set, i.e., $d = k - k' \geq 2$, then following the same argument we get $\sum_{i=1}^{k} p'_i \leq \frac{k}{d} \sum_{i=1}^{k} p_i$, and the right hand side is maximized when $d = 2$, which yields the desired upper bound on the total price of $p'$.

Concave valuation functions: Note that if the valuation function is concave, then the optimal solution $x^*$ can be derived as follows. Order the sellers in a weakly increasing order of their cost and rename them so that $c_i$ is the $i$-th smallest cost. Then, if we let $v(k) - v(k - 1)$ denote the marginal change in the buyer's value after adding a $k$-th seller to a set of $k - 1$ sellers, the optimal solution corresponds to the prefix of sellers in the aforementioned ordering for which $v(k) - v(k - 1) \geq c_k$, i.e., their marginal contribution to the buyer's value is at least as high as their cost. In any SPE equilibrium, the price offered to the winning sellers, i.e., the first k sellers in the ordering is the same, and equal to $v(k + 1) - v(k)$. It is easy to verify that these prices would support $x^*$ as a SPE equilibrium, so we now show that no other price vector can support $x^*$ as a SPE equilibrium. Consider any other price vector that supports $x^*$ as a SPE equilibrium and note that if one of the smallest $k$ prices, i.e., a price of one of the $k$ winning sellers of $x^*$, is higher than $v(k + 1) - v(k)$, then dropping that seller from the winning set would increase the buyer's utility. On the other hand, if one of these k prices is lower than $v(k + 1) - v(k)$, then the corresponding seller could raise their price while remaining in the winning set. □

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
