# OpenReview forum: "Procurement Auctions with Best and Final Offers"
_ACM.org/TheWebConf/2025/Conference — WWW 2025 Poster_

### Official Review · Reviewer_uwNC · 2024-12-02

**Novelty:** 6
**Technical Quality:** 6

**Review:**

This paper studies sequential procurement auctions with best and final offer (BAFO), where a buyer wants to acquire goods from a set of sellers that are approached sequentially and sellers have the option to freeze their price. This is a type of auction that has been used in practice but has not been extensively studied. The paper shows that such auctions achieve efficient outcomes for a broad class of buyer valuation functions but that final prices depend on the order in which sellers are approached.

**Strengths:**
+The paper is mostly clear
+The theoretical results in the paper seem novel
+The related work section is brief but seems comprehensive

**Weaknesses:**
- The paper might not be relevant to the Web Conference
- The practical implications of the paper are not clear
-The paper has many typos

**Detailed comments:**

First, I want to clarify that I am not an expert in auctions and was surprised that this paper was assigned to me. I did my best to understand the key ideas of the paper and provide constructive feedback. Still, I hope that other reviewers will lead the discussion regarding the theoretical contributions of the paper. I will expand on some of the weaknesses I have identified:

- Relevance to the Web Conference: the Web Conference does have an Economics, Online Markets and Human Computation Track with one of the topics being Advertising Auctions, Markets, and Exchanges. However, it is unclear why this paper is relevant for advertising or other Web applications. It might be a better fit for a conference such as ACM EC or WINE.

Practical implications: From a practitioner's perspective, I would like to understand the paper's practical implications. For instance, if one adopts BAFO actions versus other types of auctions, how would that benefit sellers or buyers? I would also like to know how the theoretical results presented here might guide the application of BAFO auction in a given setting compared with the more constrained settings considered in previous works.

- Writing: I have identified many typos in the paper such as "empty The auction" (page 5), "finalized" (page 2), and "restriced" (page 2). I'm sure there are many more typos not listed here. I recommend the authors to use a spellchecker to solve these.

**Questions:**

1) Why is this paper relevant to the Web conference?

2) What are the practical implications of the paper?

**Reviewer Confidence:**

1: The reviewer's evaluation is an educated guess

**Scope:**

1: The work is irrelevant to the Web

---

### Official Review · Reviewer_Jycz · 2024-12-02

**Novelty:** 7
**Technical Quality:** 7

**Review:**

This paper analyzes procurement auctions with best and final offers (BAFO). The procurement auction has n sellers (selling not necessarily substitutable goods) and 1 buyer. BAFOs allow the seller to permanently fix their price while remaining in the auction. The results characterize subgame perfect equilibria, which ensure no buyer could make a single deviation from their strategy profile and improve their utility. The main results consider sequential name-your-BAFO auctions and descending BAFO auctions. Sequential name-your-BAFO auctions sequentially ask each seller for their BAFO. Descending BAFO auctions sequentially ask each player whether they would like to lower their price by an increment or fix their current price as a BAFO. They show both auction types are efficient, in the sense that they maximize the utility of the buyer.

Overall, I enjoyed the setting considered and the results. The paper was clearly written and engaging. The focus is on not-necessarily-substitutable goods, which is very well motivated. The authors provide a compelling argument for this feature of the setting (i.e., *not* assuming the buyer’s valuation function is submodular).

My main conceptual concern about this work is the focus of the analysis on the efficiency of subgame perfect equilibria, rather than strategyproofness. SPE seems like a weak and unrealistic equilibrium concept, since in almost any situation, I imagine any player could alter several of their moves and perhaps achieve greater utility for themselves. Strategyproofness would ensure this. Perhaps SPE is justified by a computational boundedness condition? Could the authors comment on this analysis choice? With such a forgiving equilibrium concept, the fact that the BAFO auctions are efficient does not seem like a strong result.

Parts of the model could have used more motivation. For example: “the sellers know the buyer’s valuation function and the sellers’ costs (i.e., the bare minimum payment that they need to receive), but the buyer has no prior information regarding the sellers’ costs.” What situations does this describe? The authors did not provide a motivation. Maybe I should think about this as a worst-case situation (from the perspective of the buyer), so that the efficiency results can be thought of as worst-case?

Claim 5.1 is written in terms of sequentially raising prices, rather than sequentially lowering them. This seems like a mistake. Am I missing something?

A few other minor points:
    - The comparison to prior analysis is not as clear as I would have liked to see. In particular, as the authors note, there are two differences between their setting and prior work. (The differences are the BAFO structure and the fact that offers are sequential/public.) I am not sure how things would change if BAFO were implemented in a simultaneous-offer setting (beyond the inefficiency in the simultaneous chopsticks setting), and would have liked to see discussion of this point.
    - The “(utilitarian) social welfare” is just the buyer’s utility, if I understand correctly. Why are you calling it a social welfare? This confused me especially since the authors had just introduced the utilities of the *sellers*.
    - It seems like the number of interactions between buyer and seller may be large, especially if the currency increment is small. Is there a bound on the number of interactions between sellers and the buyer, other than n * h? I imagine it would be a convenience for implementation if the number of interactions scaled sublinearly with h.

This paper makes a valuable first step towards understanding sequential BAFO actions, and I think it suggests interesting directions for further analysis, especially strategyproofness results.

**Questions:**

Is the argmax in line 240 (def. of D(v; p)) always unique? If so, how? If not, does this create problems for the definition of gross substitutes property?

**Reviewer Confidence:**

4: The reviewer is certain that the evaluation is correct and very familiar with the relevant literature

**Scope:**

4: The work is relevant to the Web and to the track, and is of broad interest to the community

---

### Official Review · Reviewer_kGXz · 2024-12-02

**Novelty:** 4
**Technical Quality:** 3

**Review:**

This paper is out of the scope of the Systems track. This paper is about auction theory which is a transversal area that does not represent a contribution to Systems, which is the track where the paper has been submitted to. Due to a wrong selection of track, I am afraid I  do not have the knowledge which is so far away from my area of expertise and I cannot do a technical assessment beyond stating that this paper cannot be accepted as a "Systems" paper.

I think this paper may fit in the "Economics, online markets and human computation" track, but I am not the one who has to do that assessment. In case the paper remains in the System track I will recommend to reject because it is out of scope.

Therefore the values I am providing in technical quality is irrelevant because the problem is the scope of the paper.

**Questions:**

The only question I have is why this paper has been submitted to this track and why it was not desk rejected, or transfer to a most appropriate track.

**Reviewer Confidence:**

3: The reviewer is confident but not certain that the evaluation is correct

**Scope:**

1: The work is irrelevant to the Web

---

### Official Review · Reviewer_sZvH · 2024-12-02

**Novelty:** 3
**Technical Quality:** 3

**Review:**

1. The paper fails to motivate the problem as being relevant to the web. The work considers an alternative form of auction format that is related to economics theory and has no apparent direct relevance to the Web.

2. The paper also does not provide realistic examples of seller-buyer auctions that are similar to seller-buyer behaviors observed in the web in the context of Real Time Biddings. Instead, examples from supply chain and other industries are presented.

3. This is a theoretical framework on a variant of auction format that is an improved of earlier economical works. The literature work that has been cited also includes prior economical frameworks around descending and ascending auctions.

**Questions:**

1. I fail to see why this paper was submitted to the conference given its irrelevance to the web. In case I am missing something, please explain how it was deemed to be related to the web.

**Reviewer Confidence:**

1: The reviewer's evaluation is an educated guess

**Scope:**

1: The work is irrelevant to the Web

---

### Official Review · Reviewer_wYkG · 2024-12-04

**Novelty:** 3
**Technical Quality:** 3

**Review:**

This paper is about sequential procurement auctions, and specifically, those auctions where sellers are offered "Best and Final Offer" strategies. The authors show that sequential procurement auctions that include "BAFO" strategies are efficient, even when valuation functions exhibit substitutes/complements.

The findings on efficiency and mechanism qualities seem reasonable though I did not follow the proofs line-by-line to check for correctness. However, given the focus on computing in this conference, I wonder whether you see any relevance to algorithms, and whether the subgame perfect equilibrium strategies/efficient strategies as a function of valuation functions can be computational complex, etc? Are there any experiments, visuals, simulations, or closed-form solutions that you are able to produce as the result of the analysis?

My main issue with this paper was with clarity.
1. Could you have a place where you define "BAFO" formally?
2. Could you define the auction setting clearly and formally as a primer for the uninitiated reader?
3. It is wholly unclear to me what is meant by theta^hat as opposed to theta. The preliminaries gives the following relation theta^hat_j(s) = theta_j(s).
4. Could you add a table to the preliminaries that includes each variable and what they mean?
5. The chopstick example is helpful until the remark 3.1. Could you explain, justify, or perhaps prove it? In general, many sections end with these "remarks" in lieu of concluding or summary thoughts/discussion, and the entire paper also lacks an introduction. Normally, I think no conclusion is fine for papers, but in this case, I find myself having trouble following the take-aways or significance of the various discussions and syllogisms in the paper.

**Questions:**

Questions are reproduced from the discussion above. One new question (minor) is added at the end.

I wonder whether you see any relevance to algorithms, and whether the subgame perfect equilibrium strategies/efficient strategies as a function of valuation functions can be computational complex, etc? Are there any experiments, visuals, simulations, or closed-form solutions that you are able to produce as the result of the analysis?

My main issue with this paper was with clarity.
1. Could you have a place where you define "BAFO" formally?
2. Could you define the auction setting clearly and formally as a primer for the uninitiated reader?
3. It is wholly unclear to me what is meant by theta^hat as opposed to theta. The preliminaries gives the following relation theta^hat_j(s) = theta_j(s).
4. Could you add a table to the preliminaries that includes each variable and what they mean?
5. The chopstick example is helpful until the remark 3.1. Could you explain, justify, or perhaps prove it? In general, many sections end with these "remarks" in lieu of concluding or summary thoughts/discussion, and the entire paper also lacks an introduction. Normally, I think no conclusion is fine for papers, but in this case, I find myself having trouble following the take-aways or significance of the various discussions and syllogisms in the paper.

Additional question: Why do you assume all costs are divisible by discrete denominations? What happens when the prices can be expressed as any real number?

**Reviewer Confidence:**

2: The reviewer is willing to defend the evaluation, but it is likely that the reviewer did not understand parts of the paper

**Scope:**

3: The work is somewhat relevant to the Web and to the track, and is of narrow interest to a sub-community